# Structure of anellovirus-like particles reveal a mechanism for immune evasion

Shu-hao Liou[1,2,8], Rajendra Boggavarapu[1,8], Noah R. Cohen[1,3], Yue Zhang [1], Ishwari Sharma[1], Lynn Zeheb[1], Nidhi Mukund Acharekar[1], Hillary D. Rodgers[1], Saadman Islam[1,4], Jared Pitts[1], Cesar Arze [1], Harish Swaminathan[1,5], Nathan Yozwiak[1,6], Tuyen Ong[1], Roger J. Hajjar[1,6], Yong Chang[1], Kurt A. Swanson [1] ✉ & Simon Delagrave [1,7]

Anelloviruses are nonpathogenic viruses that comprise a major portion of the human virome. Despite being ubiquitous in the human population, anelloviruses (ANVs) remain poorly understood. Basic features of the virus, such as the identity of its capsid protein and the structure of the viral particle, have been unclear until now. Here, we use cryogenic electron microscopy to describe the first structure of an ANV-like particle. The particle, formed by 60 jelly roll domain-containing ANV capsid proteins, forms an icosahedral particle core from which spike domains extend to form a salient part of the particle surface. The spike domains come together around the 5-fold symmetry axis to form crown-like features. The base of the spike domain, the P1 subdomain, shares some sequence conservation between ANV strains while a hypervariable region, forming the P2 subdomain, is at the spike domain apex. We propose that this structure renders the particle less susceptible to antibody neutralization by hiding vulnerable conserved domains while exposing highly diverse epitopes as immunological decoys, thereby contributing to the immune evasion properties of anelloviruses. These results shed light on the structure of anelloviruses and provide a framework to understand their interactions with the immune system.

Anelloviruses, the principal constituent of the commensal human virome, are universally acquired in infancy and found throughout the body[1–3]. Since the discovery of the original torque teno virus in 1997[4], three highly diverse genera of human Anelloviridae have been described. These viruses elicit weak immune responses that permit multiple strains to co-exist and persist for years in a typical individual[5]. However, because they do not cause disease[6] and due to the lack of an in vitro culture system, the structural elements of the anellovirus capsid remain poorly understood[7,8].

Anelloviruses comprise a single-stranded, negative-sense circular DNA genome ranging from ~2.0–3.9 kilobases (kb). Among the three genera of human anelloviruses, the *Alphatorquevirus* (torque teno virus; TTV) genome is ~3.8 kb, the *Gammatorquevirus* (torque teno midi virus; TTMDV) genome is ~3.2 kb, and the *Betatorquevirus* (torque teno mini virus; TTMV) genome is ~2.9 kb. Through alternative splicing, at least six different open reading frames (ORFs) are encoded by these genomes, the longest one being ORF1[9]. A distinguishing feature of ORF1 is the presence at the N-terminus of several arginine residues

[1]Ring Therapeutics, 140 First Street Suite 300, Cambridge, MA 02139, USA. [2]Present address: Carbon Biosciences, Waltham, MA 02451, USA. [3]Present address: AbbVie Bioresearch Center, Worcester, MA 01605, USA. [4]Present address: GSK, Cambridge, MA 02139, USA. [5]Present address: DaCapo Brainscience, Cambridge, MA 02139, USA. [6]Present address: Gene and Cell Therapy Institute, Mass General Brigham, Cambridge, MA 02139, USA. [7]Present address: Delagrave Life Sciences, LLC, Sudbury, MA 01776, USA. [8]These authors contributed equally: Shu-hao Liou, Rajendra Boggavarapu. ✉e-mail: kswanson@ringtx.com

and some lysines which are consistently observed in anelloviruses of all genera. This arginine-rich motif (ARM) is seen in several viral families and is thought to function as a nuclear localization signal as well as to bind the viral genome[10,11]. ARM-containing viruses typically contain a jelly roll domain characteristic of viral capsid proteins[12]. Therefore, we hypothesized that the ORF1 protein also contains a jelly roll domain. Recently, Butkovic et al. sequenced >250 ANV genomes from Weddell seal (Leptonychotes weddellii) and grizzly bear (Ursus arctos horribilis) samples and used sequence similarity detection and structural prediction to suggest ORF1 would form a canonical jelly roll structure like those found in circular single-stranded DNA (ssDNA) viruses such as circoviruses[13]. However, no experimental evidence was available to confirm the presence of such a domain in the anellovirus ORF1. Indeed, a few failed attempts at expressing full-length ORF1 have been reported by others[9], limiting characterization of this protein.

Here we describe the expression, purification, and structural characterization of the anellovirus virus-like particle generated in two different cell lines. Initial expression of the ORF1 protein in insect cells (Sf9) suggested a proteolytically susceptible region of ORF1 that resulted in the removal of the C-terminus. With the C-terminus of the ORF1 polypeptide removed, the protein spontaneously formed symmetric 60-mer particles, as determined by cryogenic electron microscopy (cryo-EM). To generate a higher resolution structure, we generated an ORF1 polypeptide lacking the C-terminus (ΔC-Term) in human cells (Expi 293) and determined its structure. Particles from both expression systems adopt comparable folds, including an icosahedral particle core. Extending from the particle are two subdomains, herein referred to as P1 and P2, that form a spike structure around the particle's 5-fold axis. The spike domains place the P2 subdomain, comprised primarily of the hypervariable region (HVR), on the outermost surface of the particles. We therefore propose that anelloviruses have evolved this structural orientation to display their highly diverse sequences on the particle surface as a mechanism for immune evasion that potentially explains the weak immune responses to anelloviruses detected in humans.

## Results

### Anellovirus LY1 ORF1 forms virus-like particles when the C-terminus is proteolyzed in insect cells

Our initial efforts to study the structure of an ANV particle, derived from a *Betatorquevirus* isolate called LY1[14], were performed with the full-length ORF1 (residues 1–672) expressed in insect cells (Sf9) using a baculovirus system (pFastbac, Thermo Fischer) encoding a codon-optimized ORF1 construct (Fig. 1). Full-length ORF1 assembled into particles ~32 nm in diameter (Fig. 1D), similar to previously reported estimates of anellovirus size[15]. However, full-length ORF1 particles lacked the structural homogeneity and symmetry expected of viral particles. Furthermore, we noticed that ORF1 tended to degrade in the cells or during purification. Genome packaging by ARM-containing viruses is believed to occur when the positive arginine residues bind a negatively charged genome, overcoming electrostatic repulsion between ARM domains to permit capsid assembly. To prevent potential proteolysis from occurring in the ARM by trypsin-like proteases, we designed the LY1 ΔARM construct wherein residues 2–45 are deleted (Fig. 1A, B). LY1 ΔARM expression was confirmed using a rabbit polyclonal antibody raised against a spike P1 domain peptide (residues 485–502; Fig. 1B). We observed a band consistent with LY1 ΔARM by western blot above the 62 kDa marker (expected mass of 73.3 kDa; Fig. 1C). However, we continued to observe a smaller ORF1 band accumulating below the 62 kDa marker during purification, confirming the proteolysis was not limited to the ARM region. To identify the region of proteolysis, we generated polyclonal antibodies to peptides from the jelly roll domain at the extreme N-terminus (residues 46–58) and the C-terminal domain at the C-terminus (residues 635–672) of the LY1 ΔARM construct (Fig. 1B) and confirmed the presence of the

N-terminal peptide of the LY1 ΔARM fragment and the absence of the C-terminal peptide by western blot analysis (Fig. 1C).

EM analysis of the LY1 ΔARM fragment showed that the particles formed were more homogeneous and symmetric in morphology relative to the particles formed by full-length ORF1 (Fig. 1D, E). The formation of homogeneous particles following genetic removal of the N-terminus and proteolysis of the C-terminus has been observed in another jelly roll-containing virus, hepatitis E (HEV)[16]. To determine if the C-terminus of LY1 ORF1 is required for particle formation, we generated virus-like particles from construct LY1 ORF1 ΔARM ΔC-Term (Δ2-45 and Δ552–672). LY1 ORF1 ΔARM ΔC-Term produced particles of similar symmetry to LY1 ORF1 ΔARM fragment (Fig. 1F). These results suggest that proteolysis or removal of the ORF1 C-terminus improved particle formation.

### ΔC-Term virus-like particle from human cells

To generate an increased number of particles to permit higher resolution structure determination, we generated an ARM-containing ΔC-Term ORF1 construct (residues 1-609) for expression in Expi 293 cells (Fig. 2). Western blot and Coomassie-stained sodium dodecyl-sulfate polyacrylamide gel electrophoresis (SDS-PAGE) analysis of the ΔC-Term ORF1 construct purified from mammalian cells showed a principal band consistent with the expected size (Fig. 2B–D). Despite containing the N-terminal ARM region, the ΔC-Term ORF1 formed symmetrical particles similar to the ΔARM fragments produced in insect cells, suggesting the presence of the ARM region does not hinder particle formation (Fig. 2E).

### Cryo-EM structure of the ANV particle

We determined the cryo-EM structure of the virus-like particles formed by the insect cell-derived ΔARM fragment to 3.9 Å resolution from 6271 particles (Supplementary Figs. 1 and 2, Supplementary Table 1) and of the mammalian cell-derived LY1 ΔC-Term ORF1 particle using cryo-EM to 2.69 Å resolution from 22,743 particles (Fig. 3, Supplementary Figs. 3, 4 and 5, Supplementary Table 2). The ORF1 particle is formed by sixty ORF1 fragments organized in an icosahedral $T = 1$ symmetry. Despite being generated in two different cell lines and the presence of the ARM region in the mammalian virus-like particle (which is not visible in the density, suggesting flexibility) the two protomer structures are very similar (1.179 Å root mean square deviation when aligned over residues 48–562; Fig. 3E). 7 β−strands within residues 46–228 (named β−strands B to H by convention) form the majority of the canonical 8-β−strand jelly roll domain, and the eighth β−strand (strand I) is formed by residues 531-542 (Fig. 1A, B, Fig. 3F). Residues 229-530 are inserted between the H and I β−strands (herein called the H-I insertion) and form the exterior of the particle surface. Insertions between jelly roll β−strands are present in other viruses such as adeno-associated virus (AAV) and canine parvovirus (CPV)[17,18]. However, the insertions for AAV2 (228 residues) and CPV (227 residues) are between G and H β−strands while the 298-residue H-I insertion in LY1 is significantly larger. The H-I insertion extends from the jelly roll domain to form a structure herein referred to as a spike domain. The spike domain is formed by two subdomains; the spike P1 subdomain (residues 229–250 and 386–530) and the spike P2 subdomain (residues 251–385).

### Jelly roll domains form particle core

Sixty LY1 jelly roll domains form the core of the virus particle (Fig. 4). The β−strands form β−sheets which are characterized by a C-H-E-F pattern on the core's exterior and B-I-D-G pattern on the core's interior. The N-terminus of strand B is oriented to place the ARM on the interior of the core, where it is positioned to bind the viral genome. The observed C-terminal residues (545-562) extend from the C-terminus of β-strand I on the interior of the particle and thread through jelly roll

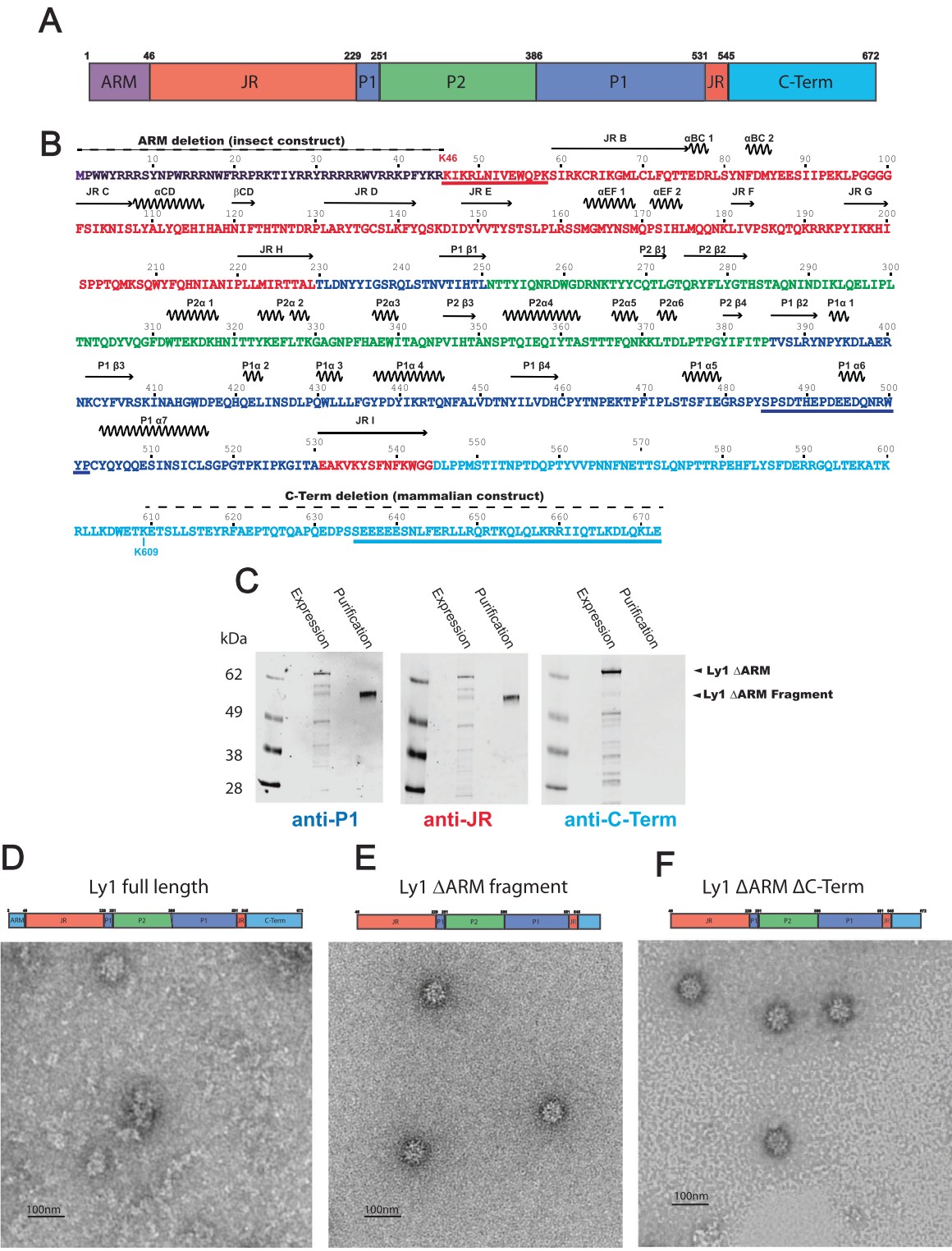

**Nature Communications** | (2024)15:7219

domains on the particle exterior near the 2-fold axis contacting the neighboring jelly roll domain (Fig. 4B). In several jelly roll-containing viruses, including the avian circovirus beak and feather disease virus (BFDV)[11], positively charged residues (arginine and lysine) oriented internally on strands B, I, D, and G are expected to bind the negatively charged viral genome (Fig. 4C). In LY1, basic residues Arg61, Lys62, Arg64 (β−strand B), Arg133 (β−strand D), Lys197 (β−strand G), Lys535, and Lys541 (β−strand I) are all oriented toward the particle interior and are likely responsible, together with the ARM motif,

for binding the negatively charged viral genome. Alignment of 2,201 *Betatorquevirus* ORF1 sequences reveals that several of these putative DNA-binding residues are conserved within ORF1 (deposited https://doi.org/10.5281/zenodo.11099132). In addition, alignment of LY1 and LY2 with 3 randomly selected *Betatorquevirus* ORF1s, 5 *Gammatorquevirus* ORF1s, and 5 *Alphatorquevirus* ORF1s shows >50% conservation of these basic residues, suggesting a general conservation for basic residues lining the interior of the ANV jelly roll particles (Supplementary Fig. 6).

**Fig. 1 | Anellovirus LY1 ΔARM construct design and proteolysis. A** A schematic representation of full-length LY1 ORF1 is shown as a cartoon labeled and colored by domains. The arginine-rich motif (ARM) is shown in purple, the jelly roll (JR) domain is shown in red, the spike P1 domain is shown in blue, the spike P2 domain is shown in green, and the C-terminal domain is shown in cyan. Residue numbers beginning each domain and the C-terminus are indicated above. **B** The sequence of full-length LY1 ORF1 colored as in A with residue numbers indicated above. A dashed line above the sequence indicates residues removed in either the insect or mammalian ORF1 constructs as indicated. Secondary structure elements are indicated above with β-strands as arrows and α-helices as zig-zag lines. The jelly roll β-strands are labeled B-I per convention while additional secondary structures are numbered by their domain. Three peptides used to generate polyclonal antibodies are underlined. **C** Western blot analysis of LY1 ΔARM after expression (Expression) and after

purification (Purification) in insect cells. A molecular weight marker is labeled to the left of the gels, while arrows on the right indicate the band of LY1 ΔARM before (LY1 ΔARM) and after proteolysis (LY1 ΔARM Fragment). Polyclonal antibodies used to probe the western blots, colored and named by the peptides used to generate them (**A**), are indicated below. **D–F** A cartoon representing the primary structure of the construct is above each micrograph, showing full-length ORF1 contains a complete ARM and C-terminal region whereas the ΔARM constructs lack the ARM region and have truncated C-terminal regions. **D** Full-length LY1 shows highly heterogeneous particles by negative staining EM. **E** LY1 ΔARM virus-like particles have a more symmetrical appearance after C-terminal proteolysis. **F** Genetic truncation of the C-terminal (Δ552-672) preserves a symmetrically structured particle.

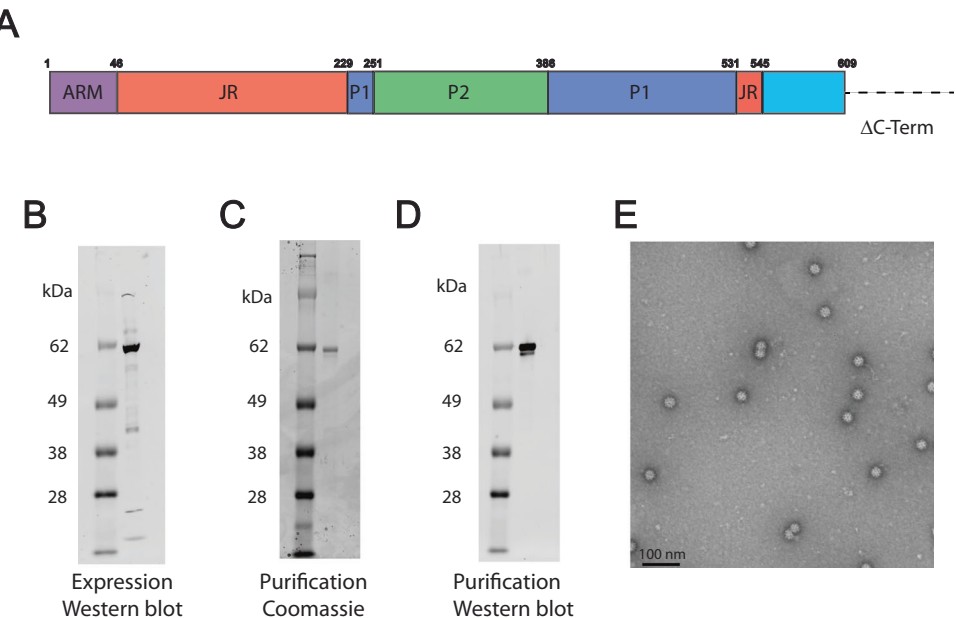

**Fig. 2 | ARM-containing ΔC-Term ORF1 forms symmetric virus-like particles in mammalian cells. A** Schematic representation of ΔC-term LY1 ORF1 is shown as a cartoon labeled and colored by domain, colored as in Fig. 1. Residue numbers beginning each domain and indicating the C-terminal deletion indicated above. **B** Western blot analysis of ΔC-term LY1 ORF1 expressed in mammalian cells.

**C** Coomassie-stained SDS-PAGE displaying the high purity of the ΔC-term LY1 ORF1. **D** Western blot analysis shows the purified ΔC-term LY1 ORF1. **E** Negatively stained electron micrograph shows the homogeneous population of ΔC-term LY1 ORF1 particles.

## Spike domain

Residues 229–530 of the H-I insertion form the spike domain that extends ~6 nm from the jelly roll core (Fig. 5). A β-strand (residues 245–250) extending from jelly roll β–strand H is the first component of the spike P1 subdomain and is N-terminal to the spike P2 subdomain (residues 251–385). The previously described HVR of ORF1 comprises the majority of the spike P2 subdomain. The remaining residues of the spike domain (residues 386–530) form five additional β–strands and eight helices, which, together with the residues 245–250 strand, fold into the spike P1 subdomain. The local resolution of P1 is only slightly lower than the jelly roll domain (~3–3.5 Å), while the resolution of P2 is within 4.5–5 Å. This is likely a consequence of both being further from the radius of gyration and some flexibility of the HVR residues.

Neighboring spike domains pack together around the five-fold symmetry axis to form a ringed structure of five spike domains henceforth called the crown (Fig. 5A, B). A receptor for anelloviruses has not been identified to date. Given the diverse tissue distribution of different anellovirus strains, it is possible that residues of the spike P2 HVR, which are the most surface-exposed, are involved in viral attachment and infection. However, we hypothesize that the hypervariable sequence of the spike P2 subdomain serves to aid in immune evasion rather than harboring a receptor-binding motif. If

this is the case, a receptor-binding motif on the better-conserved spike P1 subdomain, or even on the surface of the jelly roll core, may exist.

Absent from both structures is the majority of the C-terminal region (in the higher resolution ΔC-Term particle, we see density up to residue 562). The portion of the C-terminus we can see extends from the jelly roll core toward the surface near the 2-fold axis between the spike domain crowns. This suggests that if the C-terminus is present on the full-length viral particle, it could be located on the particle surface. The C-terminal region is predicted to be helical in nature and has a glutamate-rich region (residues 636–640 on LY1) which could N-terminally cap a conserved helical motif. To determine if the C-terminal residues of LY1 would form a helical structure, we performed circular dichroism experiments on the C-terminal peptide used to generate the aforementioned antibodies. Indeed, circular dichroism shows the C-terminal region (residues 635-672), which has a leucine-zipper motif, is helical in solution (Supplementary Fig. 7). Given that anelloviruses have evolved to evade the immune system, and that the C-terminal region is the immunodominant region of ORF1 and may be antagonistic to particle formation, we hypothesize that the helical C-terminal region is required for wild-type particle assembly but somehow modified or removed during the viral life cycle.

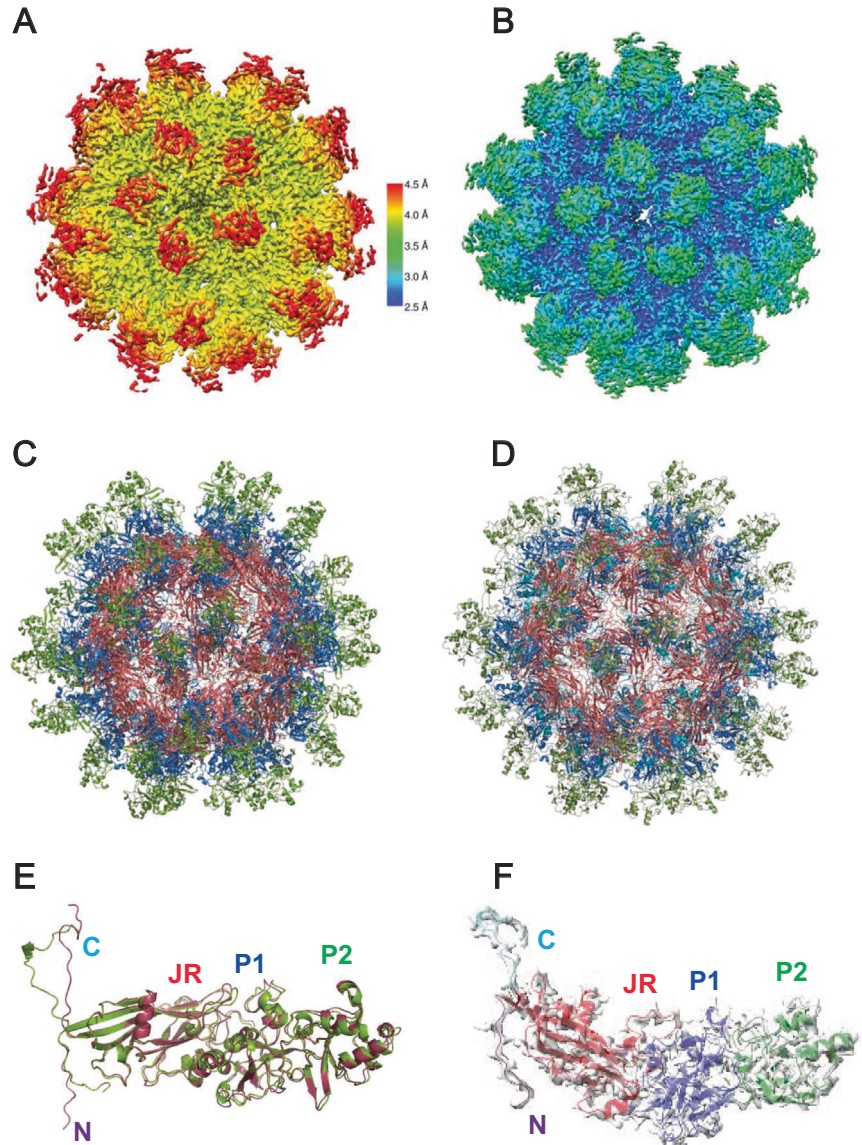

**Fig. 3 | Cryo-EM structures of ANV virus-like particles produced in insect and mammalian cells. A** The cryo-EM map demonstrating the local resolution of the SF9 cell expression purified LY1 ΔARM ORF1 particle colored by its resolution (shown on the scale to the right). **B**. The cryo-EM map marking the local resolution of expi293-expression system purified LY1 ΔC-Term particle colored by its resolution (same scale as in **A**). **C** A ribbon representation of the LY1 ΔARM 60-mer particle atomic model. **D** A ribbon representation of the LY1 ΔC-Term atomic model shown as in (**C**). **E** An overlay of the SF9 cell expression purified (red) and mammalian cell-derived (green) protomers. The observable N- and C-termini and jelly roll, P1 and P2 domains are labeled. **F** One ΔC-Term ORF1 protomer shown in its electron density with domains labeled and colored as above.

## Discussion

Despite anelloviruses constituting the majority of the human virome, their capsid structure was unknown and the identity of the capsid protein itself had not been established experimentally[19]. Our determination of the LY1 structure demonstrates that ORF1 encodes the capsid protein and that anelloviruses evolved a novel spike domain that extends around the 5-fold axis to form a crown structure. These crowns are capped with hypervariable P2 regions, which likely inhibits the development of antibodies against the better-conserved spike P1 subdomain or the jelly roll core via steric hindrance. Butkovic et al. recently performed structure prediction, corroborating the insertion of P1 and P2 subdomains between strands H and I and suggesting that anelloviruses evolved from an ancestral circovirus-like genome into which progressively larger insertions were selected[13]. Given that this insertion, containing the HVR, is the region of highest variation between the genera, it suggests the virus adopted a mechanism of

constantly mutating its crown structure as a way of evading immunodetection.

ANVs have an elongated C-terminal region, predicted to be helical, which is absent in circoviruses such as BFDV. It is unclear what role the C-terminus would play in the viral life cycle, however given that the recombinant ORF1 protein does not seem to form symmetrical particles with the C-terminal region attached, one might hypothesize it has a role in particle assembly. It is possible that manipulation of the C-terminal region through post-translational modification or interaction with some other protein factor could stimulate particle formation in the wild-type virus. Alternatively, given that the portion of the C-terminal region in the ANV-like particles is on the viral surface, perhaps the C-terminus plays a role in viral attachment and infection. Whatever the function, it is interesting that ANVs evolved this novel feature unique from the structurally similar circovirus, and this could be an area of investigation for future studies.

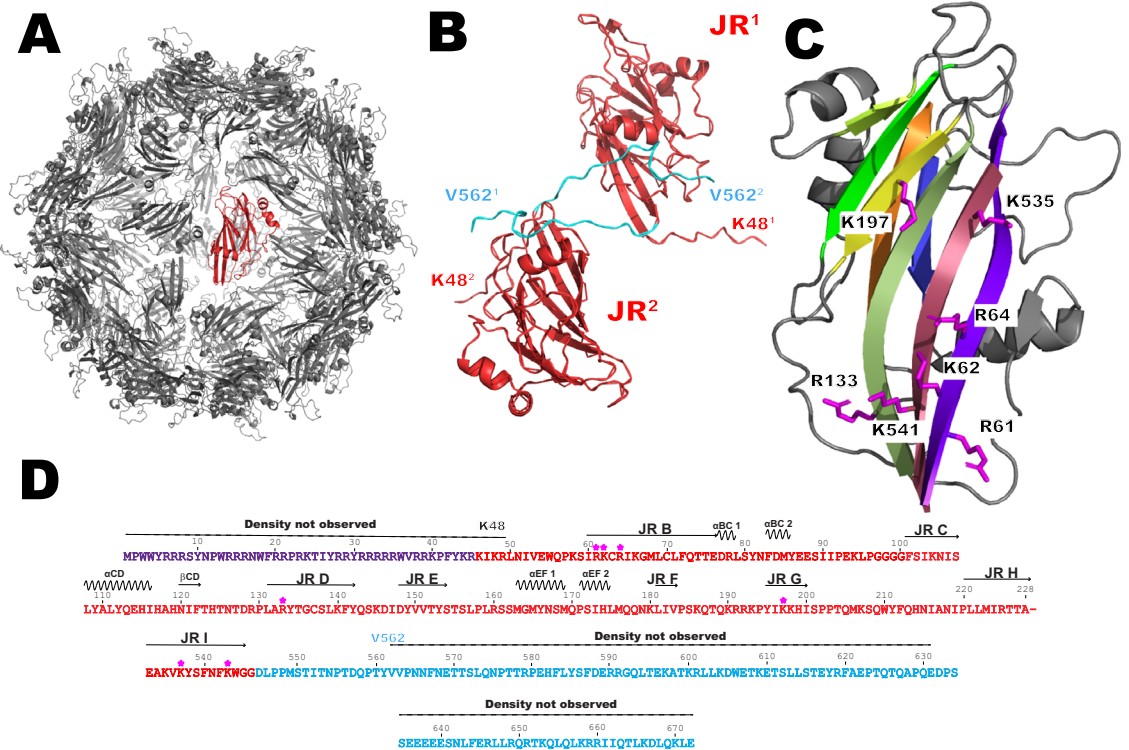

**Fig. 4 | Sixty LY1 jelly roll domains form the core of anellovirus particles. A** The LY1 core structure comprised of 60 jelly roll domains pack in icosahedral symmetry with one domain uniquely colored in red. **B** Two jelly roll domains are shown in red with the observed C-terminal domain backbone colored in cyan. The jelly roll domains are arbitrarily labeled JR[1] and JR[2] with the first (K48) and last (V562) observed residues for each protomer labeled with the corresponding number for clarity. **C** A single jelly roll domain is oriented to show the β-sheet on the interior of the particle core. Sidechains of basic residues in position to contact with the viral genome are shown and labeled. **D** The sequence of LY1 is shown for the ARM, jelly roll, and C-terminal regions. Basic residues of LY1 positioned to potentially contact the viral genome and shown in (**C**) are indicated with asterisks above the sequence along with the secondary structure elements.

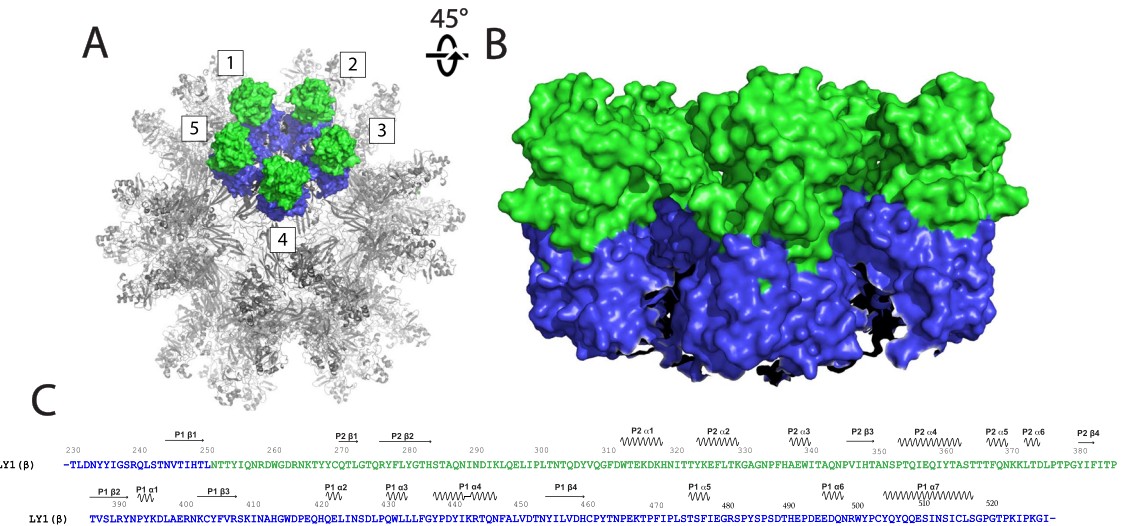

**Fig. 5 | The spike domains extend from the core on the 5-fold axis. A** The anellovirus particle structure shown as a surface rendering. The particle is shown in gray with 5 spikes forming a crown structure, numbered for clarity, shown in surface rendering and colored as in Fig. 1. **B** The exterior of the crown structure shown from the side. Five spike domains are colored as in (**A**). **C** The spike domain sequence of LY1 is shown colored as in Fig. 1. Secondary structure elements are indicated above the sequence.

While the spike domain and C-terminal regions represent novel features distinct from the otherwise structurally similar circoviruses, the ARM and jelly roll particle core are very similar. Sarker et al. have previously solved two different structural forms of BFDV capsid protein oligomers by X-ray crystallography (a decamer form and a 60 mer icosahedral capsid structure proposed to represent stages of viral maturation). When ssDNA primers were added to the decamer species, the capsomeres reassembled around the nucleic acid to form the particle, and density for ssDNA bound to basic residues of the jelly roll could be observed. ANV ORF1 particles share similar DNA contact

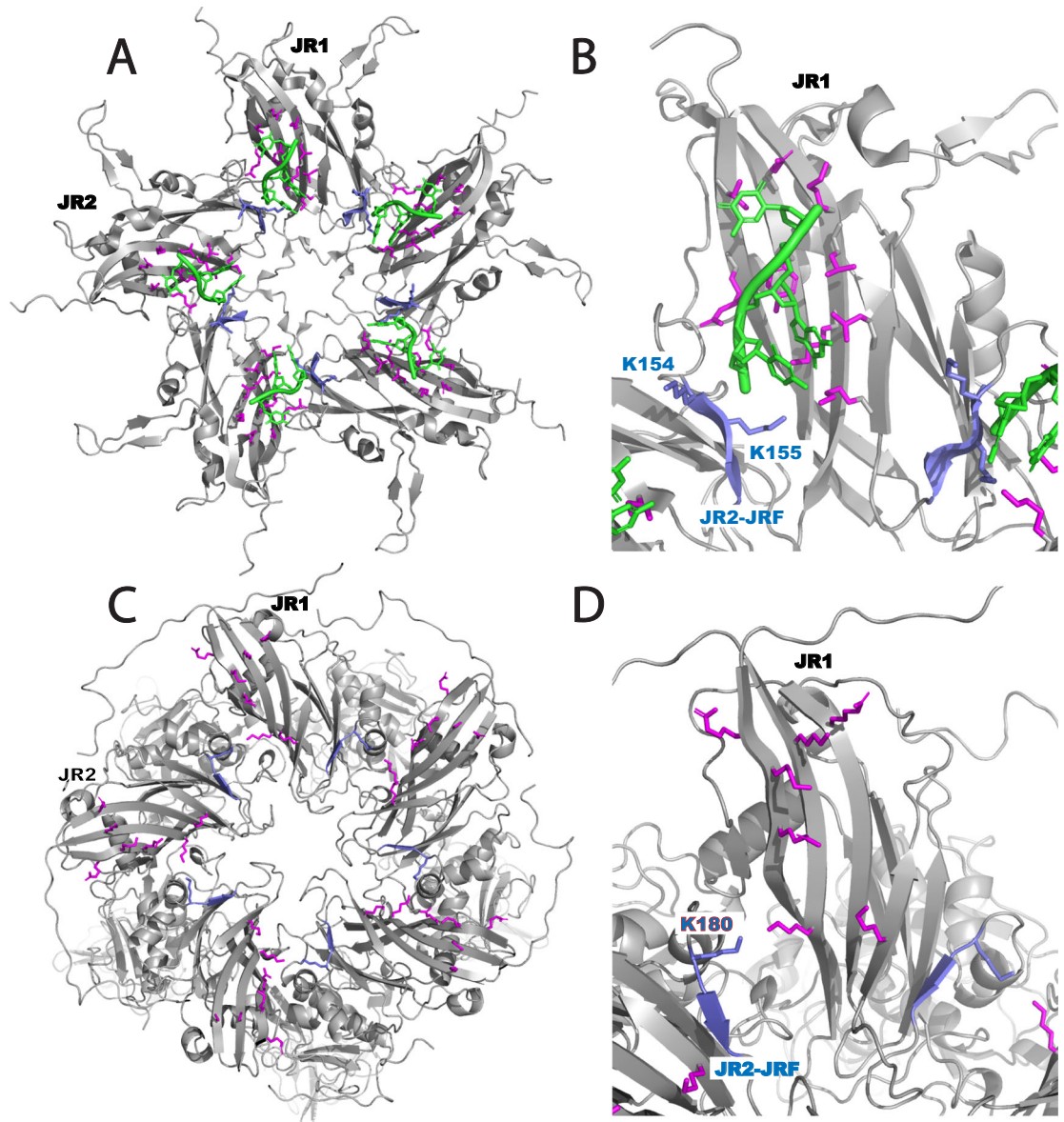

**Fig. 6 | ANV and BFDV capsids have similar DNA-interacting residues. A** A BFDV capsid pentamer showing basic residues (magenta and blue) interacting with single-stranded DNA (green). Two jelly roll domains are labeled JR1 and JR2. **B** An expanded view of (**A**) showing the F-strand of the jelly roll (blue) orients towards the neighboring jelly roll in such a way as to place basic residues Lys 154 and 155 in close proximity to the basic residues of the neighboring jelly roll (magenta). **C** An ORF1 pentamer from the ANV-like particle shown in the same orientation and labeled as the BFDV pentamer in (**A**). **D** Expanded view of (**C**) showing that residue Lys 180 and the F-strand (labeled and shown in blue) are in close proximity to basic residues of the neighboring jelly roll (magenta).

residues with BFDV capsids (Fig. 6). The B-I-D-G $\beta$-sheet represents the majority of the particle interior surface for both viruses. In BFDV, residues of jelly roll strand F are also oriented toward the particle interior (Fig. 6A, B). Two basic residues (Lys 154 and Lys 155) are oriented toward the neighboring jelly roll, placing these residues in close proximity to the basic residues of strand I and making contact with the DNA strand. While there is no visible DNA in the ANV particle structure (and no detectable DNA in the purified ORF1 sample) it is interesting to note that the jelly roll domains have a similar packing with regards to the jelly roll domains forming the pentamer (Fig. 6C, D). On LY1, basic residue Lys 180 (just N-terminal to the jelly roll F-strand) is in close proximity to basic residue Lys 535 on the I-strand of the neighboring jelly roll domain. This orientation is very similar to what is observed in BFDV. In the alignment of 2201 *Betatorquevirus* ORF1s, residues aligned to L1 Lys 180, as well as residues aligned to LY1 residues 178 and 179 are predominantly basic

(lysine, arginine or histidine) or polar (asparagine; Supplementary Fig. 8). This observation would suggest that this region could have conserved DNA-binding residues similar to the Lys residues 154 and 155 of BFDV.

The structure of the LY1 *Betatorquevirus* can now be used to guide future anellovirus research, exploring the semi-conserved regions of the P1 subdomain, C-terminal region, and jelly roll regions and how they relate to receptor binding and tissue tropism. In addition, further exploration of whether and how the ARM motif and basic residues of the jelly roll interior coordinate DNA binding and whether, like BFDV, ANVs undergo a decamer intermediate state prior to genomic encapsidation would be of interest. Lastly, the diversity and immunological stealth of anelloviruses invite continued study of their application to the delivery of therapeutic genes because they could potentially target cell types not currently addressed by existing vectors and they may be less susceptible to pre-existing immunity or to the development of

neutralizing antibodies following initial treatment[20]. In sum, the availability of the present structures will shed light on the anellovirus lifecycle and will help guide the study of anellovirus-based gene therapy vectors.

## Methods

### Antibody generation and western blot analysis

LY1 antibodies were generated by immunizing rabbits with a synthetic peptide representing one of three portions of the ORF1 protein (jelly roll residues 46–58: KIKRLNIVEWQPK, spike domain residues 485–502: SPSDTHEPDEEDQNRWYP, C-terminal domain residues 635–672: SEEEEESNLFERLLRQRTKQLQLKRRIIQTLKDLQKLE) conjugated to a carrier protein by an engineered N-terminal cysteine. Antibodies used in this study were generated and purified by Thermo Fisher Scientific, Custom Antibody Production. Briefly, rabbits were immunized twice with the indicated peptide and polyclonal antibodies were purified using protein A purification. Western blot analysis was performed using NUPAGE 4–12% gels (ThermoFisher) transferred to nitrocellulose membranes using the Transblot Turbo system (BioRad). Membranes were blocked with blocking buffer (LI-COR Biosciences), probed ~16 h with primary antibody at 1 ug/ml, and detected using anti-rabbit IRDye infrared secondary antibodies and imaging system (LI-COR Biosciences).

### Sf9 construct design, cell culture, and protein expression/purification

The LY1 ORF2 and ORF1 sequences[14] were codon-optimized for insect cells and with different ORF1 construct length variations (full-length ORF1, ΔARM with a deletion of residue 2–45, and ΔARM/ΔC-term with deletions of residues 2–45 and 552–672). The ORF2 and ORF1 constructs were cloned into the pFastBac Dual plasmid which was used to generate baculoviruses (Genscript and Medigen). To express the ORF1 proteins, Sf9 cells (Gibco 11496015) were infected by baculovirus with multiplicity of infection = 1 and the cells were cultured for 3 days at 27 °C and harvested by centrifugation. The cells were suspended in 50 mM Tris pH 8.0, 100 mM NaCl, and 2 mM MgCl₂. Cells were lysed by treatment with 0.01% Triton X-100 (Sigma-Aldrich 11332481001) and subjected to micro-fluidization, and treated with protease inhibitors (Thermo Scientific Halt Protease Inhibitor Cocktail PI78438) and DNase (Benzonase®; Sigma). Cell lysate was subsequently purified using HiTrap Heparin affinity chromatography (Cytiva) followed by size-exclusion chromatography (HiPrep 16/60 Sephacryl S-500 HR; Cytiva).

### Mammalian expression construct design, cell culture, and protein expression/purification

The LY1 ORF1 ΔC-term (1-609) sequences were codon-optimized for mammalian cells and cloned into a cytomegalovirus promoter-expression vector (Genscript). The ORF1 construct was transiently transfected into Expi293 cells (Gibco A14527) using polyethylenimine and the cells were harvested after 3 days at 37 °C by centrifugation. The cells were suspended in 50 mM Tris pH 8.0, 100 mM NaCl, and 2 mM MgCl₂. Cells were lysed by micro-fluidization and treatment with 0.01% Triton X-100 (Sigma-Aldrich 11332481001) and then treated with protease inhibitors (Thermo Scientific Halt Protease Inhibitor Cocktail PI78438) and DNase (Benzonase®; Sigma). Cell lysate was subsequently purified using HiTrap Heparin chromatography (Cytiva) followed by size-exclusion chromatography (HiPrep 16/60 Sephacryl S-500 HR; Cytiva). Lysate from 5 liters of expression was loaded onto a 100 mL Heparin Sepharose column (Cytiva, #90100192) and eluted with a 2 molar NaCl gradient followed by Capto Core 400 purification. Protein and DNA amounts were quantified by using bicinchoninic acid (Pierce™ BCA Protein Assay Kit Cat# 23225) and picogreen (Quant-iT™ Pico-Green™; the limit of detection is 50 pg) methods, respectively. Protein amount was quantified as 0.7 mg/mL and DNA estimation results show no detectable amounts of DNA. Concentrated protein quality was evaluated by SDS-PAGE, western blot, and negative stain EM.

### Negative-stained EM data collection and analysis

Purified ΔARM with a deletion of residue 2–45, and ΔARM/ΔC-term (expressed in SF9 cells) and LY1 ORF1 ΔC-term (1–609) (expressed in mammalian cells) were adsorbed for ~1 min to parlodion carbon-coated 400 mesh copper grids which were rendered hydrophilic by glow discharge at low pressure in air. Grids were washed with three drops of MilliQ water and stained with two drops of 1% uranyl acetate. Electron micrographs were recorded at a nominal magnification of 30,000 × with JEOL 1400 Flash Transmission Electron Microscope operated at 100 kV and equipped with a NanoSprint43L-MkII CMOS camera system with 43 megapixel sampling region (Advanced Microscopy Techniques).

### Cryo-EM data collection and data analysis and molecular refinement

For the SF9 fragment virus-like particle structure determination, 3 μl of 0.3 mg/ml particle sample was applied to a 1.2 × 1.3 graphene oxide grid. A total of 11,083 micrographs were collected from Glacios cryo-TEM (Thermo Fisher Scientific) operated at 200 kV with a Falcon 4 direct electron detector, at a nominal defocus range of -1.0 – -2.5 μm and accumulated dose of 19.59 e⁻/Å for a total of 15 frames in 3 min. The pixel size was 0.923 Å, with the magnification 150,000×. Automated data collection was carried out by Leginon[21] software. All micrographs were motion-corrected by Relion-4.0[22–24] implemented MotionCor2[24], and the contrast transfer function (CTF) parameters were estimated by Gctf[25]. With manually picked particles from 20 micrographs to train the network, SPHIRE-crYOLO[26] automatically picked 58,391 particles along with the PhosaurusNet network. All particles were extracted by Relion-4.0 and rescaled to 2-fold (pixel size 1.846 Å), followed by subsequence 2D classifications with 350 Å mask diameter to remove any junk particles. Two iterations of 2D classifications resulted in 11,185 particles, which were merged and reextracted to generate a de novo 3D initial model by Relion-4.0 with I1 symmetry. Notably, several similar initial models without symmetry imposed were obtained by Relion-4.0 and cisTEM[27]. To obtain a better classification result, all particles were first subjected to a Refine3D with initial angular sampling 3.7° and local angular search 0.9° per step. The alignment parameters of each particle were transferred to a 3D classification with angular sampling interval 0.9° and local angular search 5° per step. 3D classification of the entire particle set attributed most of the particles into a single class. After CtfRefine and Bayesian polishing, the post-processing results in 3.98 Å resolution under the gold standard (with Fourier shell correlation [FSC] = 0.143).

For the expi293 cell-derived ΔC-term structure determination 3 grid preparation, 3 μl of 0.4 mg/ml particle sample was applied to a 1.2 × 1.3 CCFQ and performed triple blot. A total of 19,697 micrographs were collected using a Titan Krios microscope operated at 300 kV equipped with a Gatan quantum 967 LS imaging filter, a Volta phase plate, and Gatan K3 Direct Detection camera at a nominal defocus range of −1.0 to 1.6 μm 50.53 e-/Å2 for a total of 35 frames. The pixel size was 0.834 Å with magnification 105,000×.

All micrographs were motion-corrected by Motioncor2[24] implemented in Cryosparc 3.3 and the CTF parameters were estimated by Gctf[25]. Particles were picked by automated particle picking using Cryosparc 3.3, and particles were extracted and subjected to 2D classification in Cryosparc 3.3. After iterative rounds of reference free 2D classification, 23,193 particles were selected for further processing. Initial models were generated ab initio from all selected particles, and the best 3D class was submitted to homogeneous 3D refinement that included dynamic masking. The final reported resolution of 2.69 Å was based on the gold standard FSC = 0.143 criterion[28]. Maps were visualized using Chimera[29].

The initial anellovirus TTMV-LY1 capsid monomer structure was predicted by TrRosetta[30], and the TTMV-LY1 structure was further predicted by RosettaCM[31]. Structural refinement was performed by Rosetta[32] and Phenix[33], and fine-adjusted by COOT[34,35]. Detailed definitions of the secondary elements are shown in Supplementary Fig. 9.

## Statistics and reproducibility

The negative staining-electron micrograph of full-length LY1 ORF1 (Fig. 1D) is a representative of over 100 micrographs recorded while the micrograph of C-terminal truncation (Δ552-672) is a representative of over 20 micrographs recorded (Fig. 1F). The negative staining-electron micrographs of LY1 fragment (Fig. 1E and Supplementary Fig. 1A) are representative micrographs of over 100 micrographs taken, while the cryo-electron micrograph of the same sample (Supplementary Fig. 1B) is a representative of 11,000 micrographs recorded. The negative staining-electron micrograph of ΔC-term LY1 ORF1 (Fig. 2E) is a representative of over 100 micrographs recorded while the cyro-electron micrograph of the same sample (Supplementary Fig. 3A) is a representative of 19,000 micrographs recorded.

## Sequence alignments

Sequences of select anellovirus ORF1s were taken from Genbank and aligned using Clustal Omega[36]. ORF1 sequences used in the alignment are indicated by their accession numbers in Supplementary Fig. 6. ORF1 sequences used in the alignment are indicated by their accession numbers in Supplementary Fig. 4, except LY1 (YP 007518450.1), LY2 (AGG91484.1), SAfiA (MN779270.1), and JA20 (AF122914.3). The consensus sequence is a composite of residues conserved (identity >30%) or similar residues (>50%; hydrophobic and bulky aromatic (Tyr) residues indicated with ϕ, and hydrophilic and basic residues indicated with γ). Alignments and consensus sequences are shown in Fig. 2 and Fig. 3 and Supplementary Fig. 4. Bulk alignments of 2201 Betatorquevirus ORF1 amino acid sequences retrieved from Genbank were aligned utilizing MAFFT[36] with the 'auto' parameter. Amino acid frequencies were computed for each position of the alignment and visualized in Supplementary Fig. 6.

## Circular dichroism

To determine the secondary structure of the LY1 C-terminal domain, the peptides used to generate C-terminal antibodies (residues 635–672) were analyzed at 25.8 μM in PBS on a Jasco J-815 circular dichroism spectropolarimeter using 2 mm path length cell at ambient temperature. Each data set is the average of three consecutive scans. The secondary structure was determined by using the CDPro software package[37], compared with a reference set containing 56 proteins (IBasis = 10). The final secondary structure fractions were averaged over the results from three programs (SELCON3, CDSSTR, CONTINLL) in CDPro.

## Reporting summary

Further information on research design is available in the Nature Portfolio Reporting Summary linked to this article.

## Data availability

The *Betatorquevirus* amino acid alignment referenced in this paper is available via Zenodo utilizing DOI number https://doi.org/10.5281/zenodo.11099132 (https://doi.org/10.5281/zenodo.11099133). Both particle structures from SF9- and Expi 293-expressed ORF1 were deposited to the Protein Data Bank and EMDB, and the accession codes are PDB: 8CYG, EMDB: 27077 and PDB: 8V7X, EMDB:43009 respectively. Source data are provided with this paper.

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

## Acknowledgements

We thank Dr. Joseph Che-Yen Wang for manuscript discussions and suggestions, and Maria Ericsson and the team at Harvard Medical School Electron Microscopy facility for their technical support. Cryo-EM data were collected at NanoImaging Services under the leadership of Dr. Giovanna Scapin and Dr. Phat Dip. Expi 293 produced virus-like particle cryo-EM data collection and image processing at NanoImaging Services were supported by Dr. Weili Zheng. Baculovirus was generated by Dr. Peter Pushko at Medigen. We thank Peter Riebling for proof reading and suggestions.

## Author contributions

S.-H.L. and R.B. purified samples, conducted negative EM, and processed the data with structural refinement. N.C. aided in structural model refinement. S.-H.L., J.P., and N.C. designed Sf9 ORF1 constructs and R.B. and Y.Z. designed HEK ORF1 constructs. Y.Z. and N.C. performed circular dichroism measurement. C.A. and H.S. performed multiple sequence alignments and analysis. I.S., N.M.A., H.D.R., S.I., and L.Z. performed ORF1 purification and construct screening. S.D., R.H., N.Y., T.O., and Y.C. oversaw the project. K.S. aided in ORF1 construct design and purification, aided in the design of the experiments and oversaw the structural determination. S.-H. L., R.B., S.D., and K.S. wrote the manuscript.

## Competing interests

All authors are or were employed by and hold equity interests in Ring Therapeutics. S.-H. L., N.C., R.B., Y.Z., S.D., and K.S. are the inventors on a patent application related to this work.
