## [Peer Review File · Nature Communications]

Structure of Anellovirus-like particles reveal a mechanism for immune evasion.REVIEWER COMMENTS

Reviewer #1 (Remarks to the Author):

In this manuscript, Liou, Boggavarapu and colleagues present the structure of anellovirus-like particles. The authors expressed the full-length and truncated ORF1 of betatorquevirus LY1 in insect and human cells and determined the cryo-EM structure of resultant virus-like particles. This is an important result given the prevalence of anelloviruses in various vertebrates, including humans, and the scarcity of understanding of their biology. This is the first experimentally determined structure of the anellovirus ORF1 which will be useful for the design of further experiments and interpretation of the data. Comprehensive comparative analysis of anellovirus ORF1 structural models has been recently reported (doi: 10.1093/ve/vead035) and the authors adequately acknowledge this paper in the Discussion. The manuscript is generally clearly written, but I have a few comments and questions which I would like the authors to consider.

The title is not accurate. The authors determined the structure of virus-like particles (VLP) rather than actual anellovirus virions. The title should be modified to something like "Structure of anellovirus-like particles reveals ..."

Given that N-ARM did not preclude VLP formation, the authors should state explicitly whether VLPs contained nucleic acids (e.g., cellular/plasmid RNA or DNA).

L113: The authors should be cautious stating that anellovirus particles have T=1 symmetry. Although I agree that native anellovirus particles are highly likely to have T=1 symmetry, the particles which were analyzed in this study are not virions, but VLPs. It is well known for various dsDNA and ssRNA viruses that self-assembly of the capsid proteins can produce particles of different symmetries, with some producing VLPs of smaller T compared to native virions (e.g., T=1 instead of native T=3).

L119-121: The following sentence is confusing and inaccurate. "The resulting fold of the ORF1 protomer has residues at the N- and C-termini generating the JR domain at the particle core while the intrastrand residues form the exterior of the particle surface. First, residues and the N- and C-termini do not generate the JR domain, but rather the N-ARM and C-terminal domain. Second, throughout the manuscript the authors misuse the term "intrastrand". "Intrastrand" can only mean an insertion within a strand, whereas the authors obviously mean an insertion between the strands, which would be "interstrand". The same mistake is repeated some dozen times throughout the manuscript. Furthermore, even interstrand would be confusing. Why not give this insertion a name and refer to it specifically (e.g., H-I insertion) to avoid confusion and misinterpretation?

L127 and elsewhere: "The spike domain is formed by two globular domains". A domain cannot be formed from 2 domains. Thus, P1 and P2 could be referred to as subdomains.

L142: "Alignment of anellovirus ORF1 sequences reveals that several of these putative DNA-binding residues are conserved across species, which supports their role in DNA-binding" – the authors did not perform convincing analysis of sequence conservation in either 4D or Extended Data Fig. 4. The Anelloviridae currently includes 156 species from 30 genera with many hundreds of sequenced genomes. The authors (apparently randomly) selected a handful of sequences from 3 genera for alignment. Thus, this set is hardly representative and should be presented as evidence of (general) conservation. Just a quick glance at the consensus shown in Fig. 4D raises an eyebrow – for instance, at position 97, two betatoqueviruses have a G, whereas two sequence from other genera have Y and the consensus given is G. There are many other similar instances, even for the magenta-colored positively charged residues which are presumably contacting the genome, e.g., at position 66, two closely related viruses have K, whereas other two have hydrophobic residues, and still, the consensus is given as K and colored magenta. This is misleading and unacceptable. If the authors wish to speak of conservation, they should assemble a representative sequence dataset (and explain how the sequences were chosen; be careful to have a balanced representation of different genera because this will greatly impact the conservation patterns) and redo the conservation analysis. Alternatively, they can limit the analysis to a smaller dataset and state explicitly that residues x, y, z are conserved in a particular subgroup of anelloviruses, e.g., betatorqueviruses. Otherwise, the parts on conservation (also in the section on Spike domain) should be removed and the description limited to LY1.

L170: Without prior introduction, the potential ability to bind heparin resin comes here as a surprise and irrelevant information (note that your purification procedure is explained somewhere at the very end of the manuscript and might not be read beforehand by most readers).

L174-178: I do not find the reasoning of the authors regarding the receptor-binding region being within P1 subdomain to be convincing or in any way substantiated. Why would the RBD be accessible to the receptor but not antibodies? Besides, if antibodies bound to P2, would the particle not be eliminated anyway? However, the authors present this hypothesis carefully enough and it is up to them whether to keep it.

L201: what does it mean "some sequence profile analyses"? Please be more specific and use less colloquial language. Besides, what has been suggested in the Butkovic et al paper (doi: 10.1093/ve/vead035) is that anelloviruses have evolved from a "circovirus-like" ancestor, not a circovirus. This is not the same. Please correct.

Fig. 2E: please add a scale bar.

Reviewer #2 (Remarks to the Author):

This manuscript marks the first-ever structural characterization of anelloviruses. The authors employed virus-like particles produced in both insect and human cells, leading to a comprehensive analysis of their structural features. Using cryo-electron microscopy, the researchers resolved the structures of one of the insect cell-derived constructs and the mammalian cell-derived one at resolutions of 3.9 Å and 2.8 Å, respectively. The obtained high-resolution structural data facilitated the construction of an atomic model of the capsid protein, revealing a unique architecture with a jelly roll domain and an inserted spike domain. A noteworthy aspect of the study is the proposed mechanism for immune evasion, suggesting that the particle's structure makes it less susceptible to antibody neutralization by concealing vulnerable conserved domains while exposing highly diverse epitopes on the particle surface.

The manuscript represents a significant advancement in achieving the first-ever structural characterization of anelloviruses, offering crucial insights into their biology. However, it is noteworthy to mention that the structural analysis of the constructed atomic coordinates, as well as the corresponding discussion, exhibit noticeable limitations in the manuscript. While achieving structural resolution is a commendable milestone, the manuscript could benefit from a more thorough examination of the atomic coordinates and a more in-depth discussion. A more detailed scrutiny of the atomic coordinates and a more comprehensive discussion would enhance the interpretation of the findings and strengthen the overall contribution of the manuscript to the field. Furthermore, it is pertinent to note that the presented results require a thorough review and rewriting to enhance clarity and precision in the structural analysis. Additionally, a redesign of the figures is recommended to optimize the visual representation of the findings. Authors are encouraged to consider a more in-depth and detailed discussion that comprehensively addresses the implications of the results and their significance in the broader context of anellovirus research. These suggestions aim to strengthen the quality and interpretation of the data presented in the manuscript.

As an illustrative instance, Figure 1 exhibits significant design deficiencies, making it nearly impossible to discern details within the electron microscopy images at their current size. Additionally, inconsistencies arise with the schemes over the images, where discrepancies, such as the apparent size difference in the C-terminal depicted in Panel E compared to the C-term deletion in Panel F, challenge the reliability of the visual representation. Notably, the claim that "Full-length ORF1 assembled into particles" in Figure 1D is contradicted by the predominant observation of material aggregation rather than particle assembly. A more detailed and accurate explanation is warranted to reconcile this discrepancy, enhancing clarity and transparency in the reported findings. Furthermore, the LY1 ΔARM construct lacks comprehensive biochemical data, with only a western blot analysis provided, and there is a notable absence of such data for the LY1 full-length and LY1 ΔARM ΔC-term constructs in Figure 1. Contrarily, Figure 2 presents a more detailed albeit not exhaustive analysis of LY1 ΔC-term expressed in mammalian cells. Figure 3 presents slightly different orientations of the capsids (panels A-D) and it is almost impossible to distinguish details in panels E-F. etc. etc. etc. These issues represent general shortcomings recurrent throughout the entire manuscript, warranting attention for overall improvement.

Major comments:

1. The Results section requires thorough revision and rewriting, emphasizing a detailed structural analysis rather than a purely descriptive approach to the obtained atomic coordinates. i.e. It would be beneficial to incorporate a structural comparison with other known capsid proteins from other ssDNA viruses and explore their phylogenetic implications. Additionally, a more in-depth analysis of potential structural homologies between the P1 and P2 domains with other proteins or domains is lacking. Additionally, incorporating a detailed figure illustrating the protein folding would be highly beneficial for the reader. A more comprehensive visual representation would enhance clarity and aid in understanding the structural aspects discussed in the analysis, thereby improving the overall quality of the Results section.

2. I have substantial reservations regarding the resolution achieved in the P1 and, particularly, P2 domains and the extent to which the coordinates represent predictions or have been appropriately adjusted to the density. In the main text (lines 152-154), it is mentioned, "The local resolution of P1 is only slightly lower than the JR domain ($\sim 4-4.5 \text{ \AA}$), while the resolution of P2 is within $5-6 \text{ \AA}$." However, in Figure 3, the resolution for LY1 ΔC -Term VLPs appears to be better in these domains. Upon analyzing the EM maps and coordinates provided by the authors, concerns arise regarding the adjustment, or even construction, of coordinates in certain parts of the P2 domain. I strongly urge the authors to address and clarify these points. Utilizing local reconstruction techniques may be beneficial to overcome the flexibility observed in the spike region.

3. The spike results section appears highly speculative, particularly concerning potential receptor binding surfaces and the structural data supporting immune response evasion. It may be advisable for this section to focus on an objective analysis of the experimental data, while the hypotheses related to receptor binding surfaces and immune response evasion could be more thoroughly developed in the discussion (if adequately addressed).

4. I find that the discussion in the manuscript is rather limited, especially considering the significance of the resolved structure. I would encourage the authors to develop a more comprehensive discussion that aligns with the obtained results

Reviewer #3 (Remarks to the Author):

Anellovirus Structure Reveals a Mechanism for Immune Evasion

General Comments

Dear authors, this is a quite interesting paper describing broadly the expression, purification, and structural characterization of the Anellovirus virus-like particle (VLP) generated in two different cell lines. Authors proposed that this structure renders the particle less susceptible to antibody neutralization by hiding vulnerable P1 conserved domains while exposing highly diverse epitopes (P2 domain) as immunological decoys, thereby contributing to the immune evasion properties of anelloviruses. The research is unique, soundly designed and structured where methods are appropriate and interpretations are valid. The paper is well presented, the figures are clear and well prepared. However, there is a few things that need to be addressed to improve it's worth to the broad audience of researchers interested in this non-pathogenic human virus under the family Anelloviridae. In addition discussion part need to be improved for better clarification. Please see my suggestions below:

Please see some specific comments below:

Abstract:

Abstract has been well written with proper description of objectives. Result has been explained clearly in the abstract. I would suggest to add one or two lines regarding methodology that used in this study.

Main text:

Main text has been well written with appropriate explanation of research gap on anellovirus structural biology. I have some few concerns below.

Line 29-30: Could you please rewrite the opening sentence because it is not clear to me. "The significant impact of the human virome on human physiology is beginning to emerge thanks to modern sequencing methods and bioinformatic tools." It would be great if the authors presented a relationship between human virome and structural aspects of immune evasion of anellovirus in the opening sentence of introduction.

Line 35-36: Could you please explain in introduction, what aspects of anelloviruses remain poorly understood?

Materials and methods:

In result section authors mentioned about HEK 293 cells. But, authors mentioned about Expi293 cells in the methods. Please revised the name of mammalian cells used for transfection.

Line 330: The ORF1 construct was transiently transfected into Expi293 mammalian cells (Gibco A14527) using PEI and the cells were harvested after three days at 37°C by centrifugation.

Line 100-102: ARM-containing Δ C-Term ORF1 construct (residues 1-609) for expression in HEK 293 cells.

Line 342: Please rewrite the following sentence in passive form "We used a Jeol 1200EX transmission electron microscope for screening different TTMV343 LY1 constructs."

Line 392: Please add the PDB codes in the following lines "Both VLP Structures from SF9 and HEK293 expressed ORF1 were deposited to PDB and PDB codes are XXX and YYY respectively."

Results:

Cryo-EM structure of the ANV VLP:

Line 112-114: Please check the figure number in the following lines "The anellovirus particle is formed by sixty ORF1 fragments organized in an icosahedral T=1 symmetry (Fig 2B)."

Line 137-139: Please check the figure number "In several JR-containing viruses, positively charged residues (arginine and lysine) oriented internally on strands B, I, D, and G are expected to bind the negatively charged viral genome (Fig. 3C)."

Line 139: According to Authors, basic residues Arg61, Lys62, Arg64, Lys66 (b-strand B), Arg133, Lys140 140 (b-strand D), Lys197 (b-strand G), Lys533, Lys535, and Lys541 (b-strand I) are all oriented toward the particle interior and are likely responsible, together with the ARM motif, for binding the negatively charged viral genome. Based on the sequence alignment authors claim that several of these putative DNA-binding residues are conserved across species, which supports their role in DNA-binding. However, no structural basis of capsid assembly around single-stranded DNA has been shown in the present study. Authors should explain this limitation in the discussion section.

Discussion:

Discussion part needs to be improved. Could you please discuss the possible explanation of improve virus like particle formation due to proteolysis or genetic removal of C terminal region of anellovirus.

In insect cells, Full-length ORF1 particles lacked the homogeneous symmetry expected of viral particles. Furthermore, it has been noticed that ORF1 inclined to degrade in the cells or during purification. On the other hand, the N-terminal ARM region, the Δ C-Term ORF1 formed symmetrical particles in human cells, suggesting the presence of the ARM region does not hinder particle formation. Could you please explain the possible reason for this in the discussion part. Based on the study by Butkovic et al. authors proposed that the Hyper Variable P2 domain is the region of highest variation between the genera which suggests the virus adopted a mechanism of constantly mutating its crown structure as a way of evading immunodetection. Analysis of the P1 surface shows potentially conserved patches that might have a receptor binding function, and the ever-evolving P2 domain acts to obscure these epitopes from antibody recognition. But, no experimental evidence (cell biology study) has been shown in this study. Authors claimed that anelloviruses specific cellular receptors has not been explored yet and it is also unknown that, if they exist, cellular receptors are shared between the wide varieties of strains. So, it is challenging to test this hypothesis experimentally. I would suggest to discuss the issues more extensively in the discussion section and future plan to overcome this experimental challenge.

REVIEWER COMMENTS AND RESPONSE

Reviewer #1 (Remarks to the Author):

In this manuscript, Liou, Boggavarapu and colleagues present the structure of anellovirus-like particles. The authors expressed the full-length and truncated ORF1 of betatorquevirus LY1 in insect and human cells and determined the cryo-EM structure of resultant virus-like particles. This is an important result given the prevalence of anelloviruses in various vertebrates, including humans, and the scarcity of understanding of their biology. This is the first experimentally determined structure of the anellovirus ORF1 which will be useful for the design of further experiments and interpretation of the data. Comprehensive comparative analysis of anellovirus ORF1 structural models has been recently reported (doi: 10.1093/ve/vead035) and the authors adequately acknowledge this paper in the Discussion. The manuscript is generally clearly written, but I have a few comments and questions which I would like the authors to consider.

Response:

We thank the reviewer for acknowledging the presented data is an important result. This is the first reported structure of an anellovirus (ANV)-like particle. This novel finding will undoubtedly facilitate the design of several experiments elucidating unanswered questions regarding this previously poorly-characterized anelloviruses, such as their tropism and identifying any potential receptors, identifying a receptor-binding site, understanding the roles of the novel P1/P2 subdomains and C-terminal region, etc. As the reviewer mentioned, we had acknowledged the recent reporting of an ORF1 monomer structure (as well as thorough evaluation of a likely ANV evolutionary link to a circovirus-like ancestor). However, aligned with all three reviewer's requests, we have elaborated on the findings from Butkovic *et al.* and have refocused the extended discussion to compare the structure of the ANV-like particle with that of the structurally-characterized circovirus, BFDV (below).

Reviewer 1:

The title is not accurate. The authors determined the structure of virus-like particles (VLP) rather than actual anellovirus virions. The title should be modified to something like "Structure of anellovirus-like particles reveals ..."

Response:

We thank the reviewer for this correction and have changed the title of the manuscript accordingly.

Reviewer 1:

Given that N-ARM did not preclude VLP formation, the authors should state explicitly whether VLPs contained nucleic acids (e.g., cellular/plasmid RNA or DNA).

Response:

We thank the reviewer for acknowledging this important point. We had measured the DNA content of our sample, as well as the protein concentration, and detected no DNA (LOD 50 pg). We have added that information to the methods section (line 440) and referenced the result in the extended discussion (line 265).

Reviewer 1:

L113: The authors should be cautious stating that anellovirus particles have $T=1$ symmetry. Although I agree that native anellovirus particles are highly likely to have $T=1$ symmetry, the particles which were analyzed in this study are not virions, but VLPs. It is well known for various dsDNA and ssRNA viruses that self-assembly of the capsid proteins can produce particles of different symmetries, with some producing VLPs of smaller T compared to native virions (e.g., $T=1$ instead of native $T=3$).

Response:

We appreciate the distinction and have removed the word Anellovirus from the sentence so that it states the ORF1 particle has a $T=1$ symmetry (line 124).

Reviewer 1:

L119-121: The following sentence is confusing and inaccurate.

“The resulting fold of the ORF1 protomer has residues at the N- and C-termini generating the JR domain at the particle core while the intrastrand residues form the exterior of the particle surface.

First, residues and the N- and C-termini do not generate the JR domain, but rather the N-ARM and C-terminal domain. Second, throughout the manuscript the authors misuse the term “intrastrand”. “Intrastrand” can only mean an insertion within a strand, whereas the authors obviously mean an insertion between the strands, which would be “interstrand”. The same mistake is repeated some dozen times throughout the manuscript. Furthermore, even interstrand would be confusing. Why not give this insertion a name and refer to it specifically (e.g., H-I insertion) to avoid confusion and misinterpretation?

Response:

We thank the reviewer for their comment. We have removed that sentence and re-written that paragraph (line 131) and adopted the terminology of “H-I insertion” throughout the manuscript, which aligns with the Butkovic *et al.* publication terminology.

Reviewer 1:

L127 and elsewhere: “The spike domain is formed by two globular domains”. A domain cannot be formed from 2 domains. Thus, P1 and P2 could be referred to as subdomains.

Response:

We thank the reviewer for their comment and have adopted the terminology of “subdomain” for P1 and P2 (starting line 131) and throughout the manuscript.

Reviewer 1:

L142: “Alignment of anellovirus ORF1 sequences reveals that several of these putative DNA-binding residues are conserved across species, which supports their role in DNA-binding” – the authors did not perform convincing analysis of sequence conservation in either 4D or Extended Data Fig. 4. The Anelloviridae currently includes 156 species from 30 genera with many hundreds of sequenced genomes. The authors (apparently randomly) selected a handful of sequences from 3 genera for alignment. Thus, this set is hardly representative and should be presented as evidence of (general) conservation. Just a quick glance at the consensus shown in Fig. 4D raises an eyebrow – for instance, at position 97, two betatorqueviruses have a G, whereas two sequence from other genera have Y and the consensus given is G. There are many other similar instances, even for the magenta-colored positively charged residues which are presumably contacting the genome, e.g., at position 66, two closely related viruses have K, whereas other two have hydrophobic residues, and still, the consensus is given as K and colored magenta. This is misleading and unacceptable.

If the authors wish to speak of conservation, they should assemble a representative sequence dataset (and explain how the sequences were chosen; be careful to have a balanced representation of different genera because this will greatly impact the conservation patterns) and redo the conservation analysis. Alternatively, they can limit the analysis to a smaller dataset and state explicitly that residues x, y, z are conserved in a particular subgroup of anelloviruses, e.g., Betatorqueviruses. Otherwise, the parts on conservation (also in the section on Spike domain) should be removed and the description limited to LY1.

Response:

We appreciate the reviewers concern and have removed the alignment consensus from figures 4 and 5. We did retain the fifteen ORF1 sequence alignment (Extended Data Fig. 4) and now only show the consensus from that specific alignment. We also clarify in the text that this alignment is using mostly “random” sequences (including LY1 and LY2) (line 155) and refer to this consensus as “general conservation” (line 157). In addition, we have deposited the 2201 Betatorqueviruses alignment file referenced in the original and revised text and provided a link to the file (line 555). We reference the alignment in the extended discussion with a focus on residues aligned to LY1 residues 178-180 and generated a visualization of that region (Extended Data Fig. 6).

L170: Without prior introduction, the potential ability to bind heparin resin comes here as a surprise and irrelevant information (note that your purification procedure is explained somewhere at the very end of the manuscript and might not be read beforehand by most readers).

Response:

We agree with the reviewer and have removed the reference to “potential heparin-binding” residues from the manuscript.

Reviewer 1:

L174-178: I do not find the reasoning of the authors regarding the receptor-binding region being within P1 subdomain to be convincing or in any way substantiated. Why would the RBD be accessible to the receptor but not antibodies? Besides, if antibodies bound to P2, would the particle not be eliminated anyway? However, the authors present this hypothesis carefully enough and it is up to them whether to keep it.

Response:

We appreciate the reviewer’s concern and mean to propose this as a hypothesis that needs to be further tested. We have removed the paragraph suggesting “receptor-binding patches” (former line 178, which is also aligned with comments from reviewer’s 2 and 3, below) and stated in the extended discussion that this structure can help guide future experiments pertaining to understanding receptor recognition (line 276).

Reviewer 1:

L201: what does it mean “some sequence profile analyses”? Please be more specific and use less colloquial language. Besides, what has been suggested in the Butkovic et al paper (doi: 10.1093/ve/vead035) is that anelloviruses have evolved from a “circovirus-like” ancestor, not a circovirus. This is not the same. Please correct.

Response:

We apologize for the poor summary of Butkovic *et al.* and we have further elaborated on the key findings from this paper (lines 54 and 225), which (along with the BFDV structure depicting DNA-binding residues by Sarker *et al.*) has become the new focus of our elaborated discussion (Fig. 6 and Extended Data Fig. 6).

Reviewer 1:

Fig. 2E: please add a scale bar.

Response:

We have added a scale bar to the figure.

Reviewer #2 (Remarks to the Author):

This manuscript marks the first-ever structural characterization of anelloviruses. The authors employed virus-like particles produced in both insect and human cells, leading to a comprehensive analysis of their structural features. Using cryo-electron microscopy, the researchers resolved the structures of one of the insect cell-derived constructs and the mammalian cell-derived one at resolutions of 3.9 Å and 2.8 Å, respectively. The obtained high-resolution structural data facilitated the construction of an atomic model of the capsid protein, revealing a unique architecture with a jelly roll domain and an inserted spike domain. A noteworthy aspect of the study is the proposed mechanism for immune evasion, suggesting that the particle's structure makes it less susceptible to antibody neutralization by concealing vulnerable conserved domains while exposing highly diverse epitopes on the particle surface.

The manuscript represents a significant advancement in achieving the first-ever structural characterization of anelloviruses, offering crucial insights into their biology. However, it is noteworthy to mention that the structural analysis of the constructed atomic coordinates, as well as the corresponding discussion, exhibit noticeable limitations in the manuscript. While achieving structural resolution is a commendable milestone, the manuscript could benefit from a more thorough examination of the atomic coordinates and a more in-depth discussion. A more detailed scrutiny of the atomic coordinates and a more comprehensive discussion would enhance the interpretation of the findings and strengthen the overall contribution of the manuscript to the field. Furthermore, it is pertinent to note that the presented results require a thorough review and rewriting to enhance clarity and precision in the structural analysis. Additionally, a redesign of the figures is recommended to optimize the visual representation of the findings. Authors are encouraged to consider a more in-depth and detailed discussion that comprehensively addresses the implications of the results and their significance in the broader context of anellovirus research. These suggestions aim to strengthen the quality and interpretation of the data presented in the manuscript.

Response:

We thank the reviewer for their comments regarding the significant advancement in achieving the first ANV-like particle structure. We acknowledge their comments regarding improving the quality of the structural resolution (below) as well as improving the results and discussion section (below).

Reviewer 2:

As an illustrative instance, Figure 1 exhibits significant design deficiencies, making it nearly impossible to discern details within the electron microscopy images at their current size. Additionally, inconsistencies arise with the schemes over the images, where discrepancies, such as the apparent size difference in the C-terminal depicted in Panel E compared to the C-term deletion in Panel F, challenge the reliability of the visual representation. Notably, the claim that "Full-length ORF1 assembled into particles" in Figure 1D is contradicted by the predominant observation of material aggregation rather than particle assembly. A more detailed and accurate

explanation is warranted to reconcile this discrepancy, enhancing clarity and transparency in the reported findings. Furthermore, the LY1 Δ ARM construct lacks comprehensive biochemical data, with only a western blot analysis provided, and there is a notable absence of such data for the LY1 full-length and LY1 Δ ARM Δ C-term constructs in Figure 1. Contrarily, Figure 2 presents a more detailed albeit not exhaustive analysis of LY1 Δ C-term expressed in mammalian cells. Figure 3 presents slightly different orientations of the capsids (panels A-D) and it is almost impossible to distinguish details in panels E-F. etc. etc. etc. These issues represent general shortcomings recurrent throughout the entire manuscript, warranting attention for overall improvement.

Response:

We thank the reviewer for their comments regarding the quality of the figure panels, with an emphasis on Fig. 1. We have repeated the negative stain images in figure 1 (the Sf9 material) and we have regenerated the panels of the three SF9 negative staining EMs to illustrate a better population of the sample. We have also increased the panels size to improve visibility of both the EM micrograph and the primary structure cartoon, which indicate the start and end residues of each construct (where known). The difference in the C-term length (as well as the lacking ARM motif) depicted in the three negative staining panels is meant to reflect the relative lengths (for the C-terminal region). We have further clarified this point in the legend. The starting and ending residues, depicted in Figs 1 and 2, are also noted throughout the text.

We have similarly gone through the panels of each figure in an attempt to provide consistent orientations. We agree that further elucidation in the proteolysis of the C-term domain is warranted, however the scope of the work was to determine the first ANV-like particle structure to aid future experiments (many of which are ongoing) elucidating the role of the C-terminus, as well as defining any receptor-binding sites. We have clarified that statement in the extended discussion (line 275).

Reviewer 2:

Major comments:

1. The Results section requires thorough revision and rewriting, emphasizing a detailed structural analysis rather than a purely descriptive approach to the obtained atomic coordinates. i.e. It would be beneficial to incorporate a structural comparison with other known capsid proteins from other ssDNA viruses and explore their phylogenetic implications. Additionally, a more in-depth analysis of potential structural homologies between the P1 and P2 domains with other proteins or domains is lacking. Additionally, incorporating a detailed figure illustrating the protein folding would be highly beneficial for the reader. A more comprehensive visual representation would enhance clarity and aid in understanding the structural aspects discussed in the analysis, thereby improving the overall quality of the Results section.

Response:

We appreciate the reviewer's comments and have greatly added to the discussion section. We have included a more descriptive summary of Butkovic *et. al*, in which the authors describe in detail how ANVs likely evolved from a circovirus-like ancestor with the addition of the P1 and P2 subdomains, which are absent in the circovirus family. However, to improve the discussion section we did both a structural comparison of the beak and feather disease virus (BFDV) and its well-defined DNA-binding residues and the basic residues observed in the ANV-like structure (already discussed in Fig 4 for the interior jelly roll sheet). In addition, Sarker *et. al* had identified additional basic residues (Lys 154 and 155) on the jelly roll F-strand which orient toward the neighboring jelly roll domain that also makes DNA-contact residues in BFDV. As mentioned above for reviewer 1, we reviewed our alignment of 2201 *Betatorquevirus* ORF1s and looked for conserved basic residues in this region and found that residues aligned to LY1 position 178, 179 and 180 were predominantly basic residues (Lys representing ~50% of each position) and specifically residue 180 of LY1 is a lysine residue which is oriented toward the neighboring jelly roll domain similar to BFDV K154 and K155. Additional analysis can be done comparing the ORF1 alignments and the structure and are part of future experiments suggested in the expanded discussion, however are beyond the scope of this manuscript.

Reviewer 2:

2. I have substantial reservations regarding the resolution achieved in the P1 and, particularly, P2 domains and the extent to which the coordinates represent predictions or have been appropriately adjusted to the density. In the main text (lines 152-154), it is mentioned, "The local resolution of P1 is only slightly lower than the JR domain (~4-4.5 Å), while the resolution of P2 is within 5-6 Å." However, in Figure 3, the resolution for LY1 ΔC-Term VLPs appears to be better in these domains. Upon analyzing the EM maps and coordinates provided by the authors, concerns arise regarding the adjustment, or even construction, of coordinates in certain parts of the P2 domain. I strongly urge the authors to address and clarify these points. Utilizing local reconstruction techniques may be beneficial to overcome the flexibility observed in the spike region.

Response:

We acknowledge the reviewers concerns regarding the density of the spike region (especially in the P2 subdomain) and we took their suggestion to explore local reconstruction. We did the focus classification and focused motion correction to improve the map quality. These approaches significantly improved the map quality, as can be seen with the new validation statistics (below). Resolution improved from 2.8 Å to 2.69 Å.

However, the cryo-EM density for the P2 domain remained unclear even after focus classification for both structures delta ARM LY1 and LY1 delta C term, indicating that the P2 domain is quite flexible. When we built the model, we applied different low-pass filters to the 3D density map to facilitate modeling the flexible regions using the structural envelope. Even by doing so, there are still some regions that are not covered by the cryo-EM density, resulting in CaBLAM and CA Geometry outliers as shown in the table. We attribute this to likely flexibility in the region, however sufficient density is available to build the model in the spike domain's apex. For the reviewer's consideration here we are attaching the comparative statistics of model in the previous submission and refined for the revision. We are also attaching a figure here.

Original model:

Clash score	9.47	74 th percentile
Poor rotamers	1	0.21%
Favored rotamers	465	97.69%
Ramachandran outliers	4	0.78%
Ramachandran favored	477	92.98%
Rama distribution Z-score	-1.36 ± 0.38	
MolProbity score	1.95	
Cβ deviations	1	0.20%
Bad bonds:	0 / 4348	0.00%
Bad angles:	1 / 5923	0.02%
CaBLAM outliers	22	4.3%
CA Geometry outliers	6	1.17%

Refined model

Clash score	3.6	97 th percentile
Poor rotamers	0	0.00%
Favored rotamers	472	99.16%
Ramachandran outliers	472	99.16%
Ramachandran favored	508	99.03%
Rama distribution Z-score	-1.16 ± 0.34	
MolProbity score	1.15	
Cβ deviations	0	0.00%
Bad bonds:	0 / 4348	0.00%
Bad angles:	1 / 5923	0.02%
CaBLAM outliers	4	0.8%
CA Geometry outliers	2	0.39%

A)

B)

Figure: A) Model fit into the map used in the original submission. Data was processed by regular motion correction in Cryosparc 4.0 B) Model fit into the map after revision. Data was processed using reference-based motion correction and model was built by applying

low pass filter for P2 region. The overall Density of the map improved from 2.8 Å to 2.69 Å. The new density maps have been provided to Nature Communications for review, if desired.

Reviewer 2:

3. The spike results section appears highly speculative, particularly concerning potential receptor binding surfaces and the structural data supporting immune response evasion. It may be advisable for this section to focus on an objective analysis of the experimental data, while the hypotheses related to receptor binding surfaces and immune response evasion could be more thoroughly developed in the discussion (if adequately addressed).

Response:

We appreciate the point from both reviewer 1 and reviewer 2 suggesting the prediction of an unknown receptor based on the structure and poorly conserved residues is highly speculative. We have removed that paragraph from the manuscript suggesting receptor-binding residues and simply indicated that the structure (along with further sequence analysis) can be used as tools for further investigation. Instead, we indicate that further elucidation of any receptor-binding motif will be greatly aided by our structural (and sequence alignment) analysis and changed the focus of the discussion to structural similarities with the circovirus BFDV per the reviewer's comments above.

Reviewer 2:

4. I find that the discussion in the manuscript is rather limited, especially considering the significance of the resolved structure. I would encourage the authors to develop a more comprehensive discussion that aligns with the obtained results

Response:

We appreciate the reviewer's comments and have removed the receptor-binding speculation and replaced it with the comparison of the evolutionarily-similar circovirus and its known DNA-binding residues (Figure 6 and Extended Data Figure 6) and how this observation relates to the previously published study of ANV and circovirus evolution and the structural analysis of circovirus BFDV. This new area of discussion is less speculative than the receptor-binding conservation and the role of the C-terminal regions, which we highlight will be areas of future investigation aided greatly by our structural (and sequence alignment) analysis.

Reviewer #3 (Remarks to the Author):

Anellovirus Structure Reveals a Mechanism for Immune Evasion

General Comments

Dear authors, this is a quite interesting paper describing broadly the expression, purification,

*and structural characterization of the Anellovirus virus-like particle (VLP) generated in two different cell lines. Authors proposed that this structure renders the particle less susceptible to antibody neutralization by hiding vulnerable P1 conserved domains while exposing highly diverse epitopes (P2 domain) as immunological decoys, thereby contributing to the immune evasion properties of anelloviruses. The research is unique, soundly designed and structured where methods are appropriate and interpretations are valid. The paper is well presented, the figures are clear and well prepared. However, there is a few things that need to be addressed to improve it's worth to the broad audience of researchers interested in this non-pathogenic human virus under the family Anelloviridae. In addition discussion part need to be improved for better clarification. Please see my suggestions below:
Please see some specific comments below:*

Abstract:

Abstract has been well written with proper description of objectives. Result has been explained clearly in the abstract. I would suggest to add one or two lines regarding methodology that used in this study.

Response:

We thank the reviewer for acknowledging the unique nature of our findings and that the manuscript was generally well written. We have included a statement that we are using cryo-EM to study the structure of our ANV-like particles in the abstract as suggested (line 20). We appreciate their concerns with specific aspects of the results, discussion and methods sections and have addressed specific concerns below.

Main text:

Main text has been well written with appropriate explanation of research gap on anellovirus structural biology. I have some few concerns below.

Line 29-30: Could you please rewrite the opening sentence because it is not clear to me. "The significant impact of the human virome on human physiology is beginning to emerge thanks to modern sequencing methods and bioinformatic tools." It would be great if the authors presented a relationship between human virome and structural aspects of immune evasion of anellovirus in the opening sentence of introduction.

Response:

We appreciate the reviewer's comment and have simply removed the sentence and focused on the observations described in this manuscript.

Reviewer 3:

Line 35-36: Could you please explain in introduction, what aspects of anelloviruses remain poorly understood?

Response:

We have clarified line 42 to indicate that the structural elements of the anellovirus capsid protein remains poorly understood. In addition, in line 59 we indicated that, although Alphafold modeling by Butkovic et al. suggested ORF1 was a jelly roll-containing protein we clarify that, due to previous inability to express ORF1, it was previously impossible to confirm the model experimentally. In addition, the published model is that of a monomer and our elaborated result section now demonstrates structurally-related similarities between ANV and BFDV (circoviruses and ANVs having been suggested to be evolutionarily-related; Fig. 6).

Reviewer 3:

Materials and methods:

In result section authors mentioned about HEK 293 cells. But, authors mentioned about Expi293 cells in the methods. Please revised the name of mammalian cells used for transfection.

Line 330: The ORF1 construct was transiently transfected into Expi293 mammalian cells (Gibco A14527) using PEI and the cells were harvested after three days at 37°C by centrifugation.

Line 100-102: ARM-containing ΔC -Term ORF1 construct (residues 1-609) for expression in HEK 293 cells.

Response:

We clarified in method section (line 429) and throughout the text that the mammalian cells used for expression were the Expi293 expression cell line (Gibco A14527).

Reviewer 3:

Line 342: Please rewrite the following sentence in passive form “We used a Jeol 1200EX transmission electron microscope for screening different TTMV343 LY1 constructs.”

Response:

We have revised this paragraph to reflect the new figure data and it was written in the passive tense.

Reviewer 3:

Line 392: Please add the PDB codes in the following lines “Both VLP Structures from SF9 and HEK293 expressed ORF1 were deposited to PDB and PDB codes are XXX and YYY respectively.”

Response:

We thank the reviewer for this observation. The PDB files were deposited and validated and the assigned PDB numbers are indicated in the text.

Reviewer 3:

Results:

Cryo-EM structure of the ANV VLP:

Line 112-114: Please check the figure number in the following lines “The anellovirus particle is formed by sixty ORF1 fragments organized in an icosahedral T=1 symmetry (Fig 2B).”

Response:

We do not have a specific figure for this statement and have therefore removed the figure reference.

Reviewer 3:

Line 137-139: Please check the figure number “In several JR-containing viruses, positively charged residues (arginine and lysine) oriented internally on strands B, I, D, and G are expected to bind the negatively charged viral genome (Fig. 3C).”

Line 139: According to Authors, basic residues Arg61, Lys62, Arg64, Lys66 (b-strand B), Arg133, Lys140 140 (b-strand D), Lys197 (b-strand G), Lys533, Lys535, and Lys541 (b-strand I) are all oriented toward the particle interior and are likely responsible, together with the ARM motif, for binding the negatively charged viral genome. Based on the sequence alignment authors claim that several of these putative DNA-binding residues are conserved across species, which supports their role in DNA-binding. However, no structural basis of capsid assembly around single-stranded DNA has been shown in the present study. Authors should explain this limitation in the discussion section.

Response:

We have greatly changed this paragraph in alignment with these comments and comments from reviewer 1 (above), and we have also corrected the figure reference to read Fig. 4 and Extended Data Fig. 4. In addition, we added Fig. 6. In alignment with comments from reviewer 2 to compare ANV to a similar ssDNA virus (we selected BFDV for which ssDNA contact residues were identified) to further support the hypothesis that these conserved basic residues are likely evolutionarily conserved DNA-binding (Fig. 6). However, we also indicate in the enhanced discussion that confirmation of the DNA-binding residues will be a future endeavor enabled by this structural (and sequence alignment).

Reviewer 3:

Discussion:

Discussion part needs to be improved. Could you please discuss the possible explanation of improve virus like particle formation due to proteolysis or genetic removal of C terminal region

of anellovirus.

In insect cells, Full-length ORF1 particles lacked the homogeneous symmetry expected of viral particles. Furthermore, it has been noticed that ORF1 inclined to degrade in the cells or during purification. On the other hand, the N-terminal ARM region, the Δ C-Term ORF1 formed symmetrical particles in human cells, suggesting the presence of the ARM region does not hinder particle formation. Could you please explain the possible reason for this in the discussion part.

Based on the study by Butkovic et al. authors proposed that the Hyper Variable P2 domain is the region of highest variation between the genera which suggests the virus adopted a mechanism of constantly mutating its crown structure as a way of evading immunodetection. Analysis of the P1 surface shows potentially conserved patches that might have a receptor binding function, and the ever-evolving P2 domain acts to obscure these epitopes from antibody recognition. But, no experimental evidence (cell biology study) has been shown in this study. Authors claimed that anelloviruses specific cellular receptors has not been explored yet and it is also unknown that, if they exist, cellular receptors are shared between the wide varieties of strains. So, it is challenging to test this hypothesis experimentally. I would suggest to discuss the issues more extensively in the discussion section and future plan to overcome this experimental challenge.

Response:

We appreciate reviewer's 1, 2 and 3 all pointing out that our prediction of a receptor-binding site on P1 subdomain was premature and would require further analysis. We have therefore removed this section. In addition, while we have ongoing efforts to describe the role of the C-terminal region (which is unique from the related circovirus as indicated in line 232), experimental evaluation clarifying the role of the novel ANV C-terminus is beyond the scope of this current manuscript. We therefore simply indicate in the extended results section that this structure, as well as the sequence analysis, can be used as tools for elucidating some of these questions as more experimental data becomes available. To further bolster the discussion section, we instead did a structural comparison of the related BFDV particle with well-defined DNA-contact residues with the generally conserved basic residues on ANV in similar positions.

REVIEWERS' COMMENTS

Reviewer #1 (Remarks to the Author):

The authors have adequately addressed all of my comments. I spotted a few typos, which the authors could correct during the proofs stage, if the manuscript is accepted for publication:

L206: "virtual life cycle" > "viral life cycle".

L233: "Ly1 180" > "Lys 180".

Check usage "jelly roll" versus "jellyroll". Should be uniform.

Ref 13 is a duplication of ref 21. The latter is correct, with full bibliographic information.

Reviewer #2 (Remarks to the Author):

As indicated in the previous review, this manuscript represents a significant advancement in achieving the first-ever structural characterization of anelloviruses, offering crucial insights into their biology. Authors have used cryo-EM to resolve the structure of anellovirus VLPs produced in insect and human cells. The main criticism of previous review was related to the limitations on the structural analysis and discussion of the constructed atomic coordinates. The authors have performed a notable effort to improve the manuscript and provide a level of structural analysis and discussion that, not being outstanding, meets the minimum quality standards for publication. It is also notable the improvement in the quality of the maps and specifically in the spike density that reinforce the interpretation of the density and the built coordinates. However, some changes are still required before publishing:

pp3 ll82. the statement "particles lacked the homogeneous symmetry expected of viral particles" could be more accurately phrased. The authors likely intended to refer to "structural homogeneity" or the "isometric nature" of the particles rather than "homogeneous symmetry."

pp4 ll105-106. "These results suggest that proteolysis or removal of the ORF1 C-terminus improved particle formation." Both deltaARM constructs (with and without C-term) renders more homogeneous particles than full length. However, authors only focus in the removal of the C-terminus. In this point of the results (before next section), it is not possible to know if the removal of the ARM or the removal of the C-term is the key factor for this result (which actually is discussed in the next section).

In their response letter authors state: "We have similarly gone through the panels of each figure in an attempt to provide consistent orientations." However, orientations in Figures 3 to 5 seems to be the same than previously. In the case of Figure 3 panels still present slightly different orientation of the particles and it is difficult to distinguish any detail in panels E-F.

Consider using commas as thousands separators to facilitate readability of large numbers (e.g., particle numbers in 3D reconstructions).

The legends for panels C and D in Figure 3 should be revised. Instead of referring to "3D reconstructions," they should specify that these panels depict atomic models or ribbon representations of the atomic model.

It would be beneficial to include a supplementary figure showing the monomer of the atomic model with secondary structure elements (SSEs) indicated. This would enhance readers' understanding of the structural features.

The deposited DOI (doi:10.5281/zenodo.11099132) appears to be unavailable.

Authors should ensure that maps and PDB files are deposited in the corresponding databases (EMDB and PDB), and access codes should be included in the manuscript.

Reviewer #3 (Remarks to the Author):

Dear Authors

Thank you for your responses. I appreciate your efforts in further analysis and correction. I have no additional comments.

Final Reviewer Comments:

Reviewer #1 (Remarks to the Author):

The authors have adequately addressed all of my comments. I spotted a few typos, which the authors could correct during the proofs stage, if the manuscript is accepted for publication:

Comment: L206: “virtual life cycle” > “viral life cycle”.

Response: Completed

Comment: L233: “Ly1 180” > “Lys 180”.

Response: Completed

Comment: Check usage “jelly roll” versus “jellyroll”. Should be uniform.

Response: Completed

Comment: Ref 13 is a duplication of ref 21. The latter is correct, with full bibliographic information.

Response: Completed

Reviewer #2 (Remarks to the Author):

As indicated in the previous review, this manuscript represents a significant advancement in achieving the first-ever structural characterization of anelloviruses, offering crucial insights into their biology. Authors have used cryo-EM to resolve the structure of anellovirus VLPs produced in insect and human cells. The main criticism of previous review was related to the limitations on the structural analysis and discussion of the constructed atomic coordinates. The authors have performed a notable effort to improve the manuscript and provide a level of structural analysis and discussion that, not being outstanding, meets the minimum quality standards for publication. It is also notable the improvement in the quality of the maps and specifically in the spike density that reinforce the interpretation of the density and the built coordinates. However, some changes are still required before publishing:

Comment:

pp3 ll82. The statement “particles lacked the homogeneous symmetry expected of viral particles” could be more accurately phrased. The authors likely intended to refer to “structural homogeneity” or the “isometric nature” of the particles rather than “homogeneous symmetry.”

Response: We have changed the sentence to read “However, full-length ORF1 particles lacked the structural homogeneity and symmetry expected of viral particles.”

pp4 ll105-106. “These results suggest that proteolysis or removal of the ORF1 C-terminus improved particle formation.” Both deltaARM constructs (with and without C-term) renders more homogeneous particles than full length. However, authors only focus in the removal of the C-terminus. In this point of the results (before next section), it is not possible to know if the removal of the ARM or the removal of the C-term is the key factor for this result (which actually is discussed in the next section).

Response: We appreciate what the reviewer is suggesting, but as they point out the observed symmetry is improved in both ARM-containing and ARM-deleted constructs as indicated later in

the text, showing the C-term clipping/removal is a key variable. As the other two reviewers did not express concern over the text focusing on the C-terminus at this point in the manuscript, we are hoping to leave this text as-is to keep the flow of the manuscript. We are open to suggestions from the editor.

Comment: In their response letter authors state: "We have similarly gone through the panels of each figure in an attempt to provide consistent orientations." However, orientations in Figures 3 to 5 seems to be the same than previously. In the case of Figure 3 panels still present slightly different orientation of the particles and it is difficult to distinguish any detail in panels E-F.

Response: We now understand what the reviewer is requesting. The orientation of the structures various panels were selected to highlight the point being made. For example, Figure 4A shows the jelly roll from the particle exterior where Figure 4C shows the jelly roll as viewed by the interior of the particle as indicated in the legend. As the other two reviewers did not express concern, would like to keep these orientations as they are and are open to the editor's comments. We have added a Supporting Data Fig. 9 with several panels showing the secondary structure details of the particle to provide more details per the reviewer's comment here and below.

Comment: Consider using commas as thousands separators to facilitate readability of large numbers (e.g., particle numbers in 3D reconstructions).

Response: Completed

Comment: The legends for panels C and D in Figure 3 should be revised. Instead of referring to "3D reconstructions," they should specify that these panels depict atomic models or ribbon representations of the atomic model"

Response: We have changed the legend to read:

"Fig. 3. Cryo-EM structures of ANV virus-like particles produced in insect and mammalian cells. A. The cryo-EM map demonstrating the local resolution of the SF9 cell expression purified LY1 Δ ARM ORF1 particle colored by its resolution (shown on the scale to the right). B. The cryo-EM map marking the local resolution of expi293-expression system purified LY1 Δ C-Term particle colored by its resolution (same scale as in A). C. A ribbon representation of the LY1 Δ ARM 60-mer particle atomic model. D. A ribbon representation of the LY1 Δ C-Term atomic model shown as in C. E. An overlay of the SF9 cell expression purified (red) and mammalian cell-derived (green) protomers. The observable N- and C-termini and jelly roll, P1 and P2 domains are labeled. F. One Δ C-Term ORF1 protomer shown in its electron density with domains labeled and colored as above."

Comment: It would be beneficial to include a supplementary figure showing the monomer of the atomic model with secondary structure elements (SSEs) indicated. This would enhance readers' understanding of the structural features.

We have added a Supporting Data Fig. 9 with several panels showing the secondary structure details of the particle to provide more details per the reviewer's comment here and above.

Comment: The deposited DOI (doi:10.5281/zenodo.11099132) appears to be unavailable.

Response: The doi is active and has been downloaded by two people (perhaps the other two reviewers). We have added the http address (line 503) to make access easier for the reader.

Authors should ensure that maps and PDB files are deposited in the corresponding databases (EMDB and PDB), and access codes should be included in the manuscript.

Response: The PDB and EMDB access codes are indicate in the manuscript (line 352-353).

Reviewer #3 (Remarks to the Author):

Dear Authors

Thank you for your responses. I appreciate your efforts in further analysis and correction. I have no additional comments.

Response: We thank all of the reviewers for their constructive criticism.